# Rituximab-Containing Risk-Adapted Treatment Strategy in Nodular Lymphocyte Predominant Hodgkin Lymphoma: 7-Years Follow-Up

**DOI:** 10.3390/cancers13081760

**Published:** 2021-04-07

**Authors:** Novella Pugliese, Marco Picardi, Roberta Della Pepa, Claudia Giordano, Francesco Muriano, Aldo Leone, Giuseppe Delle Cave, Alessandro D’Ambrosio, Violetta Marafioti, Maria Gabriella Rascato, Daniela Russo, Massimo Mascolo, Fabrizio Pane

**Affiliations:** 1Department of Clinical Medicine and Surgery, Hematology Section, University of Naples “Federico II”, Via Sergio Pansini, 5, 80131 Naples, Italy; marco.picardi@unina.it (M.P.); robertadellapepa@gmail.com (R.D.P.); claudiagiordano91@hotmail.com (C.G.); francesco.muriano@gmail.com (F.M.); aldo.leone93@libero.it (A.L.); giusedellecave87@gmail.com (G.D.C.); alexander.dambrosio@gmail.com (A.D.); violettamar@hotmail.it (V.M.); mg.rascato@gmail.com (M.G.R.); fabrizio.pane@unina.it (F.P.); 2Pathology Unit, Department of Advanced Biomedical Sciences, University of Naples “Federico II”, Via Sergio Pansini, 5, 80131 Naples, Italy; daniela.russo@unina.it (D.R.); massimo.mascolo@unina.it (M.M.)

**Keywords:** nodular lymphocyte predominant hodgkin lymphoma, rituximab, progression-free survival

## Abstract

**Simple Summary:**

The current literature on NLPHL therapy is scarce due to the disease rarity. Our study aims to focus on the management and treatment strategies of NLPHL based on risk stratification. This paper contributes to the current literature, based mainly on retrospective studies and on small cohort studies, confirming the benefit of Rituximab for patients with NLPHL, in particular for those with advanced disease. We hope that our results can be confirmed by larger cohort studies from the research community.

**Abstract:**

Background: Nodular lymphocyte predominant Hodgkin lymphoma (NLPHL) is a rare variant of HL that accounts for 5% of all HL cases. The expression of CD20 on neoplastic lymphocytes provides a suitable target for novel treatments based on Rituximab. Due to its rarity, consolidated and widely accepted treatment guidelines are still lacking for this disease. Methods: Between 1 December 2007 and 28 February 2018, sixteen consecutive newly diagnosed adult patients with NLPHL received Rituximab (induction ± maintenance)-based therapy, according to the baseline risk of German Hodgkin Study Group prognostic score system. The treatment efficacy and safety of the Rituximab-group were compared to those of a historical cohort of 12 patients with NLPHL who received Doxorubicin, Bleomycin, Vinblastine, Dacarbazine (ABVD) chemotherapy followed by radiotherapy (RT), if needed, according to a similar baseline risk. The primary outcome was progression-free survival (PFS) and secondary outcomes were overall survival (OS) and side-effects (according to the Common Terminology Criteria for Adverse Events, v4.03). Results: After a 7-year follow-up (range, 1–11 years), PFS was 100% for patients treated with the Rituximab-containing regimen versus 66% for patients of the historical cohort (*p* = 0.036). Four patients in the latter group showed insufficient response to therapy. The PFS for early favorable and early unfavorable NLPHLs was similar between treatment groups, while a better PFS was recorded for advanced-stages treated with the Rituximab-containing regimen. The OS was similar for the two treatment groups. Short- and long-term side-effects were more frequently observed in the historical cohort. Grade ≥3 neutropenia was more frequent in the historical cohort compared with the Rituximab-group (58.3% vs. 18.7%, respectively; *p* = 0.03). Long-term non-hematological toxicities were observed more frequently in the historical cohort. Conclusion: Our results confirm the value of Rituximab in NLPHL therapy and show that Rituximab (single-agent) induction and maintenance in a limited-stage, or Rituximab with ABVD only in the presence of risk factors, give excellent results while sparing cytotoxic agent- and/or RT-related damage. Furthermore, Rituximab inclusion in advanced-stage therapeutic strategy seems to improve PFS compared to conventional chemo-radiotherapy.

## 1. Introduction

Nodular lymphocyte predominant Hodgkin lymphoma (NLPHL) is a rare HL subtype since it represents approximately 5% of all cases [1]. Although NLPHL is characterized by multiple and late relapses, it is acknowledged as having a more indolent course and potentially a better prognosis compared with classical HL (cHL) [2,3].

Treatment choice is based on the disease stage and may include radiation therapy (RT) with or without chemotherapy (CT), borrowing approaches and strategies from cHL [4]. The current National Comprehensive Cancer Network [4] and European Society for Medical Oncology [5] guidelines recommend using RT alone for first-line therapy of patients with non-bulky early stage NLPHL, and CT using an ABVD (Doxorubicin, Bleomycin, Vinblastine, Dacarbazine) [6] or CHOP (Cyclophosphamide, Doxorubicin, Vincristine and oral Prednisolone) regimen plus RT for most advanced-stages. Considering possible recurrences requiring further therapies in NLPHL, a less toxic treatment schedule would be desirable in order to reduce treatment-related mortality and morbidity.

From this consideration, the different phenotypes of expression on the cell of origin (“popcorn” cells or lymphocyte-predominant (LP) cells in NLPHL is a feature of CD20 and CD79a expression, unlike Hodgkin-Reed-Sternberg cHL cells), provides the rationale for the inclusion of the anti-CD20 antibody Rituximab in the treatment schemes of this disease, as a low toxic component able to replace CT and/or RT in order to reduce observed toxicity. Rituximab use is widely consolidated in the treatment of B-cell non-Hodgkin lymphoma, while its role in the treatment schedule for all stages of NLPHL is still not clear. Anti-CD20 antibody treatment with Rituximab showed impressive activity in relapsed NLPHL, as proven by studies from the German Hodgkin Study Group (GHSG) and the Stanford group [7,8,9]. More recently, the GHSG conducted a single-arm phase 2 study to evaluate efficacy and safety of Rituximab as single agent in patients with newly diagnosed stage I-A NLPHL [10]. Little is known about the role and optimal schedule of the administration of anti-CD20 antibodies in the induction treatment of NLPHL in all the different stages.

To shed more light on the best front-line treatment for NLPHL, we conducted a retrospective study in patients with this entity who were referred to our institution for therapy based on Rituximab (Rituximab-containing group) between 1 December 2007 and 28 February 2018. We compared the progression-free survival (PFS) and overall survival (OS) of these patients with those of a historical cohort of NLPHL patients, previously referred to our institution for therapy based on combined modality treatment (anthracycline-containing CT plus RT) or on only anthracycline-containing CT during the period between 1 October 2001 and 30 November 2007. In addition, great attention was paid to the occurrence of therapy-related side-effects. These data were partially previously described [11]; here, we report an increasing number of patients treated in the Rituximab-containing group with a longer follow-up.

## 2. Results

### 2.1. Patients Characteristics

The demographic and clinical characteristics of the analyzed patients are reported in Table 1; the relative frequencies of patient features in the two groups were compared and were similar for adverse prognostic factors. Risk-adapted strategies for historical cohort and Rituximab-containing group are summarized in Table 2 and described in the Materials and Methods section.

### 2.2. Outcome Measures and Survival

Overall, 27 of the 28 enrolled patients obtained a complete remission (CR) to first-line therapy. One patient in the historical cohort showed insufficient response to front-line therapy and was considered as primary refractory.

The median follow-up for the entire study population was 7 years (range 1–11 years) and did not differ between the historical cohort (median follow-up 7.7 years, range 4–11 years) and Rituximab-containing group (median follow-up 6.6 years, range 1–11 years; *p* = 0.14).

During the follow-up period, one relapse in the early stage sub-set and two relapses in the advanced-stage sub-set were observed in the historical cohort, while none of the patients among the Rituximab-containing group showed relapses. Thus, NLPHL patients treated with Rituximab-containing risk-adapted protocol showed a better 7-year PFS compared with historical cohort (100% vs. 66.7, respectively; (95% confidence interval—CI 44.7–100), *p* = 0.036; Figure 1a). In particular, among the advanced-stage patients, the median PFS was 23.8 months for patients of the historical cohort compared with median PFS (not reached) for patients treated according to the Rituximab-containing protocol (*p* = 0.002, Figure 1b). In contrast, no differences in terms of PFS between treatment protocols were found for early favorable (*p* = 0.62, Figure 1c) and early unfavorable (*p* = 1, Figure 1d) patients.

The OS was similar for the two treatment groups, with only one event in the Rituximab-containing group (*p* = 0.39, Figure 2). This patient had documented CR at the one-month assessment after the last dose of scheduled immunochemotherapy but died due to bleomycin-induced pneumonitis.

Short- and long-term side-effects (according to the Common Terminology Criteria for Adverse Events—CTCAE, version 4.03) were more frequently observed in the historical cohort. Grade 3–4 neutropenia was more frequent in the historical cohort compared with the Rituximab-containing group (58.3% vs. 18.7%, respectively; *p* = 0.03). Long-term non-hematological toxicities were more frequently observed in the historical cohort. Among the 12 cases, six suffered from thyroid disease, two from lung fibrosis, two from avascular necrosis of the femoral head, and one from valvular heart disease (most likely related to irradiation). One patient in the rituximab group died after induction immunochemotherapy due to bleomycin-induced pneumonitis.

## 3. Discussion

In this study, the outcomes of two consecutive series of patients with NLPHL, one comprising 12 patients (historical cohort) and the other including 16 patients, were compared. The first series received an induction treatment with either classical combined modality treatment, i.e., a CT/RT combination, or a full-course of CT alone. The second series received front-line Rituximab-containing risk-adapted treatment. All of the 16 patients (100%) who received Rituximab-containing treatment achieved CR at restaging, while one out of the 12 patients who were treated with the classical strategy (ABVD ± RT) showed refractory disease. At diagnosis, this patient had advanced-stage disease with right cervical, obturator and iliac nodal localizations and three risk factors according to the International Prognostic Score (IPS) (fair risk) [13].

Despite the more indolent clinical behavior of NLPHL and the more favorable prognosis, particularly for patients with early stage disease, compared to cHL, late and occasionally multiple relapses are frequently observed [2,14,15]. Therefore, there is a need for breakthrough treatment advances that, along with low toxicity, demonstrate greater efficacy in improving overall prognosis. Due to its rarity, there is no wide consensus as regards induction therapy for NLPHL.

Outcomes from the principal studies reported in the setting of early stage and advance-stage NLPHLs are summarized in Table 3 and Table 4.

For patients with stage I-II NLPHL, RT, either alone or in combination with anthracycline-containing CT, has been the mainstay of treatment, and several studies have reported very good outcomes, with the 10-year PFS being >80% [4,18,20,21], similar to that reported in our series for both treatment groups.

Advani et al. reported the results of a phase II study of Rituximab induction with or without Rituximab maintenance in both previously treated and untreated NLPHL. In the setting of 21 newly diagnosed patients, the estimated 5-year PFS was 41.7% vs. 51.9% for patients treated with Rituximab induction vs. Rituximab induction + maintenance, respectively (median PFS 1.9 vs. 5.6 years, *p* = 0.37), with a trend for better PFS for patients with stages I–II compared with stage III (median PFS 5.8 vs. 1.8 years, respectively) [12]. In this trial, however, the early stage group also included patients with risk factors. Our results underline the importance of therapy stratification on the basis of risk at baseline and indicate that the patients with risk factors deserve CT, while a prolonged exposure to Rituximab is sufficient to guarantee long lasting protection against disease recurrence for those patients with very limited stage disease. In fact, in our series, we restricted the use of Rituximab as a single agent only to the patients at stages I and II without risk factors. However, in order to reduce the possibility of relapses, we prolonged the anti-CD20 antibody treatment with further eight infusions (once every three months), distributed over two-year maintenance therapy. Therefore, as seen in indolent B-cell non-Hodgkin lymphomas, the use of Rituximab maintenance seems to either delay or avoid relapses, with the PFS being similar for patients with early favorable NLPHL treated either with combined CT/RT modalities or the Rituximab-containing regimen (Figure 1d). This similarity in PFS was confirmed after a long follow-up: the median follow-up for early stage Rituximab-free-treated patients was 6.5 years with a range of 3.2–11 years, and 6.25 years with a range of 5.1–11 years for early favorable patients undergoing Rituximab induction + maintenance.

In a phase II trial from the GHSG, investigating Rituximab monotherapy in newly diagnosed stage I-A NLPHL, the 4-year PFS was 77.1% [10]. Although Rituximab was shown to be an active and safe single agent in the treatment of NLPHL, the authors remarked the better PFS reported for patients with early stage disease treated with RT (5-year PFS 95%) [2,20] or CT [21], suggesting that Rituximab alone is less effective than RT or CT. In this study, the role of Rituximab maintenance was not explored and despite an initial impressive overall response rate (ORR) of 100% with CR observed in 24 of the 28 patients, the short time frame Rituximab exposure appears to be insufficient to preclude the likelihood of continuous early relapses, as indicated by PFS rates of 96.4%, 85.3% and 81.4% at 12, 24, and 36 months, respectively, and the 25% total disease recurrences observed in the study. The difference in terms of PFS from our study is likely to be related to the results from the use of maintenance Rituximab in our cohort. Secondary malignancies were the most frequent cause of death among patients treated with RT [20] (Table 3 and Table 4). However, in this cohort of 28 patients described by Eichenauer et al. [10] and treated only with Rituximab, two patients, nevertheless, developed secondary malignancies after 2 and 4 years, respectively, from the initial NLPHL diagnosis. RT-related long-term iatrogenic complications have led to increased exploration of different treatment modalities.

In particular, in order to spare RT-related toxicities, Pinnix et al. [25], in a cohort of 71 NLPHL stage I/II reported no differences in tumor control among patients treated with extended-field RT (EFRT), involved-field RT (IFRT) and involved-site RT. Similar results were obtained in GHSG clinical trials in which EFRT was compared with IFRT, leading the authors to conclude that IFRT is associated with less toxicity and should be preferred over EFRT [22].

In contrast, Savage et al. reported—in an era-to-era comparison—a better 10-year PFS for the ABVD regimen (91%) compared with RT alone (65%) for the treatment of early stage NLPHL [21]. In a more recent retrospective multicenter analysis on 559 NLPHLs, RT and combined RT/CT treatment showed similar 5-year PFS (91.1% and 90.5%, respectively), without differences in terms of OS [29].

Taken together, the already available data and results of our study indicate that front-line Rituximab treatment, when followed by a two-year maintenance therapy, seems to have a low toxicity and a high efficacy in treating patients with early stage NLPHL without risk factors. These data need to be confirmed in a larger study population cohort at a time when there is more evidence for the use of limited field RT.

In the remaining patients of our series, according to a risk-adapted treatment approach, we combined Rituximab therapy to polychemotherapy, giving a total of four cycles of ABVD to the patients at stage I and II but with one or more risk factors (early unfavorable) and six cycles of ABVD to those patients who presented advanced-stage (III or IV) disease. Importantly, the inclusion of Rituximab in early unfavorable cases did not modify the PFS, but spared patients from RT-related toxicity (Table 3).

Overall, Rituximab was well tolerated in our patients and severe toxic side-effects were not observed when administered in combination with polychemotherapy, except for one patient: a 69 year old, with advanced-stage disease, who died in complete remission after immunochemotherapy due to pneumonitis as a consequence of a bleomycin agent.

Defining the optimal treatment regimen for patients with NLPHL in advanced-stage is even more challenging [37], and the suitable treatment options include either a B-cell lymphoma directed CT (CVP (Cyclophosphamide, Vincristine and oral Prednisolone) or CHOP or bendamustine) [33,34], or ABVD plus or minus Rituximab as valid options [2,20]. Although the use of the R-CHOP-21 regimen seems to be associated with a lower risk of transformation as late as 20 years after the initial diagnosis [38], the number of patients treated with R-CHOP is modest (Table 4). Thus, further observations are needed to confirm this impression. In relation to ABVD, R-CHOP causes more cases of peripheral neuropathy (with vincristine compared to vinblastine), has a higher cumulative steroid exposure, and a potentially greater likelihood of infertility (cyclophosphamide vs. dacarbazine). In contrast, ABVD results in more pulmonary toxicity (bleomycin) and is more difficult to deliver to elderly patients. In summary, R-CHOP-21 may be a highly effective treatment of patients with advanced-stage NLPHL, but is not compared prospectively to ABVD [33]. On the other hand, a preliminary study showed that R-bendamustine is a promising candidate for the therapy of NLPHL, however, the results of this study should be verified in larger prospective trials to confirm the favorable risk-benefit profile of this regimen [34].

In our setting of advanced-stage NLPHL, the introduction of Rituximab appears to provide long-term protection from relapses, as compared with the classical CT/RT approach. Indeed, approximately all of the advanced-stage patients treated with ABVD ± RT and in CR after completion of therapy, relapsed before the end of follow-up. The already described trend toward a continuous pattern of relapse after classical treatment [39] was particularly marked in our patients with advanced-stage presentation. On the other hand, almost all of the patients who received Rituximab-containing therapy remained in CR 7 years after initiation of treatment. These results underscore the role of the anti-CD20 antibody in NLPHL advanced-stage therapy.

Among the seven previously untreated patients with advanced-stage NLPHL who received Rituximab induction ± Rituximab maintenance, reported by Advani et al. [12], despite the ORR of 100%, the median PFS of 22 months achieved in this setting, suggests a short-lived response with Rituximab monotherapy, independent from Rituximab maintenance, and the necessity to use CT [4]. Our results in the setting of advanced-stage disease confirm the importance of CT and suggest an additional role of Rituximab in delaying relapse and durable response.

However, all these treatment modalities are associated with a well-known increased risk of toxicity. In our advanced-stage Rituximab group, one death was recorded due to pneumonitis and the patient was 69 years old, thus, in our opinion, the bleomycin agent could be spared in older patients. In a ^18^F-fluoro-deoxy-glucose (FDG)-positron emission tomography (PET)/computed tomography (CT)-2 oriented scenario, it would be possible to omit bleomycin or reduce the CT agents of the ABVD scheme, when Rituximab is employed, especially in an elderly population, in which the risk of bleomycin induced pneumonitis and overall infection rates seems to be higher. On the other hand, Eichenauer and Engert recently, in *blood* 2020, recommend treating patients with advanced-stage NLPHL with a more aggressive therapeutic regimen, i.e., four to six cycles of BEACOPPesc (escalated bleomycin, etoposide, doxorubicin, cyclophosphamide, vincristine, procarbazine, and prednisone) ± RT [40].

Furthermore, recently, Eichenauer et al. [36], in a PET-2-guided escalated program for advanced stage NLPHL, compared outcomes of PET-2 positive patients after two cycles of chemotherapy, who were randomized to receive six/four additional cycles of eBEACOPP or six additional cycles of eBEACOPP plus rituximab. No differences in terms of 5-year PFS were recorded between the two cohorts (19 patients receiving eBEACOPP (67.5%) and 12 patients receiving additional rituximab (73.3%)). Thus, in PET-2 guided scenarios, Rituximab implementation does not appear to improve outcome, these data need to be further investigated in a larger cohort of patients [36].

Cumulatively, these data suggest that single-agent Rituximab is not appropriate for the treatment of newly diagnosed advanced stage patients as responses are not durable, but it is reasonable to combine rituximab with chemotherapy, including ABVD, to achieve better PFS. The primary limitation of our study is the potential for confounding in a retrospective analysis. It is possible that some uncontrolled variables (interim metabolic imaging, such as FDG-PET/CT scans, RT technique, dose, and fractionation, and/or supportive therapy) could have influenced both treatment utilization and survival in this series, including treatment bias. However, the treatment choice was only influenced by the time-period of the first diagnosis and by stage stratification, thus, reducing the effect of confounding variable. The small number of patients, especially when stratified according to stage and risk factors, represents another limitation of this study, and does not allow us to draw determining conclusions. In the context of this rare disease, it is difficult to conduct studies, although retrospective, on larger cohorts of well-characterized patients, however, data from registries or pooled data from larger study groups or institutions could clarify the role of Rituximab in the first-line treatment of NLPHL. Nevertheless, the long period in which our patients were followed strengthens the reported data. Given the rarity of NLPHL subtype, we believe that it is unlikely that randomized trials will be conducted in this setting. As a result, we believe that retrospective and well-balanced analyses, such as the current one, can serve an important role in identifying disparities in care and improving patient outcomes.

## 4. Materials and Methods

Clinical records and outcome of 28 consecutive patients with recent diagnosis of NLPHL followed at the Hematology Unit of the Federico II University of Naples (Italy) were retrospectively analyzed. This analysis compares patients with NLPHL treated with two different risk-adapted strategies over different periods.

The study was approved by the local ethics committee and written informed consent was obtained from the patients.

Lymph node biopsy information reported morphologic, and immunohistochemical features for all analyzed patients. NLPHL histologic diagnosis was performed according to the Rey classification before and after 2008 [41], according to the World Health Organization (WHO) 2008 [42] and 2016 classification [43]. For all patients, lymph node samples obtained by excisional biopsies were available, routinely fixed in formalin and embedded in paraffin (histologic sections stained with hematoxylin and eosin, and Giemsa). All biopsies were retrospectively assessed by two expert hematopathologists with more than 10 years of experience with hematopathological analysis (M.M and D.R.) to confirm the initial NLPHL diagnosis and CD20 expression on neoplastic cells.

All the analyzed patients showed a performance status ≤3 and were eligible to receive immuno/chemotherapy and/or RT. Patients had a complete anamnesis and underwent physical examination, routine laboratory tests, total-body FDG-PET/CT and bone marrow biopsy; thus, before treatment they were carefully staged according to the Ann-Arbor classification.

Patients were further stratified according to the GHSG in early favorable, early unfavorable and advanced-stage [44] as follows: early favorable, were patients in clinical stage I–II without risk factors; early unfavorable, were patients with clinical stage I–II with at least 1 risk factor among mediastinal mass larger than one-third of the maximum thoracic width; extra-nodal disease; elevated erythrocyte sedimentation rate (ESR) > 50 mm/h without B symptoms or ESR > 30 mm/h with B symptoms (B symptoms: fever, night sweats, unexplained weight loss >10% over 6 months); involvement of ≥3 nodal areas. Patients with advanced-stage disease were considered to be those in clinical stage IIB with large mediastinal mass and/or extranodal disease, and clinical stages III–IV (Table 2).

The International Prognostic Score was applied to better categorize patients with advanced-stage NLPHL [13].

### 4.1. Treatment

#### 4.1.1. Historical Cohort Risk-Adapted Strategy

Between 1 October 2001 and 30 November 2007, 12 consecutive patients with NLPHL (historical cohort) were treated, according to their stage, with conventional ABVD regimen (day 1 and day 15 of each cycle of 4 weeks) plus IFRT [6,45]. In particular, all patients in stage I/II received ABVD schedule for four cycles followed by IFRT (30 Gy), while patients in stage III/IV received six cycles of ABVD (Table 2).

#### 4.1.2. Rituximab-Containing Risk-Adapted Strategy

Between 1 December 2007 and 28 February 2018, 16 consecutive adult patients with NLPHL received Rituximab (375 mg/m^2^) intravenous infusion alone or combined with ABVD according to the baseline risk as follows: patients with early favorable disease received Rituximab as a single agent, once per week for four weeks, followed by Rituximab maintenance (once every three months for 2 years); patients with early unfavorable disease received Rituximab once per month plus 4 ABVD cycles; patients with advanced-stage disease received Rituximab two times per month (on day 1 and 15) for 6 months plus 6 ABVD cycles (Table 2).

### 4.2. Outcome Measures

The treatment efficacy was assessed after completion according to the 2007 Revised Response Criteria for Malignant Lymphoma [46]. Patients classified as complete responders (disappearance of all evidence of disease) were monitored every 3 months for 24 months and thereafter every 6 months, using clinical-laboratoristic examinations, ultrasonography scans and chest radiography [47]. Patients’ follow-up was updated in April 2020.

In the historical cohort, response to treatment was assessed 3 months after the last dose of RT in patients who underwent combined induction treatment with ABVD plus RT, or one month after the last dose of chemotherapy in patients who underwent chemotherapy alone. In the rituximab group, response to treatment was assessed one month after the last dose of rituximab induction (early favorable disease) or one month after the last dose of immunochemotherapy (early unfavorable and advanced-stages).

Short- and long-term treatment related adverse events were recorded and classified according to the CTCAE, v4.03.

### 4.3. Statistical Considerations

The primary endpoint of this study was to compare the PFS of these two different risk-adapted strategies for the treatment of NLPHL. PFS was defined as the period between treatment start and relapse, progressive disease or the last follow-up visit if none of these events occurred. Secondary endpoints were the OS—defined as the time from diagnosis and the last follow-up visit or death for any cause—and the treatment-related adverse event occurrence and severity.

The Kaplan–Meier method was used to calculate survival; comparisons were made by using the log-rank test. Baseline characteristics and incidence of adverse events were compared using the χ^2^ test; Student’s *t*-test was also used. All statistical analyses were performed by using *R*-software, version 3.6.

## 5. Conclusions

In conclusion, taken together, the already available data and the results of our study indicate that front-line Rituximab induction treatment, when followed by two-year Rituximab maintenance, seems to have a low toxicity and a high efficacy in treating patients with early stage NLPHL without risk factors, replacing previous chemotherapy and/or RT schemes, and sparing the toxicity derived from the use of chemotherapy and/or RT. Furthermore, the combination of anti-CD20 antibodies with the classical ABVD schedule within a risk-adjusted treatment strategy may be used in the therapeutic approach of patients with NLPHL with more aggressive/extensive disease, showing a significant advantage in terms of PFS for advanced-stages.

The small number of patients, especially when these are stratified according to stage and risk factors, constitutes the main limitation of this study, thus, our results need to be confirmed in larger cohort prospective studies.

## Figures and Tables

**Figure 1 cancers-13-01760-f001:**
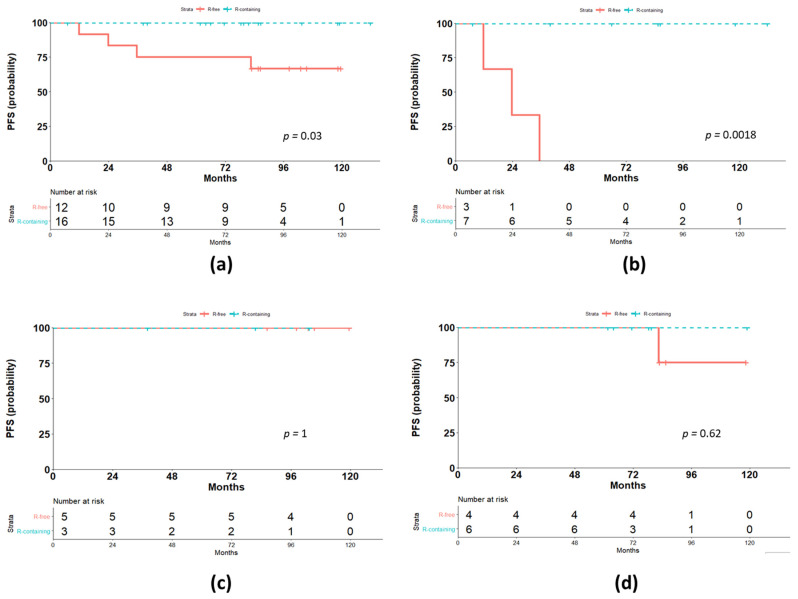
Outcome according to the treatment protocol. (**a**) Comparison of progression-free survival (PFS) for the two treatment groups, Rituximab-containing (R-containing) and Rituximab-free (R-free); (**b**) comparison of PFS for advanced-stages; (**c**) comparison of PFS for early unfavorable stages; (**d**) comparison of PFS for early favorable stages.

**Figure 2 cancers-13-01760-f002:**
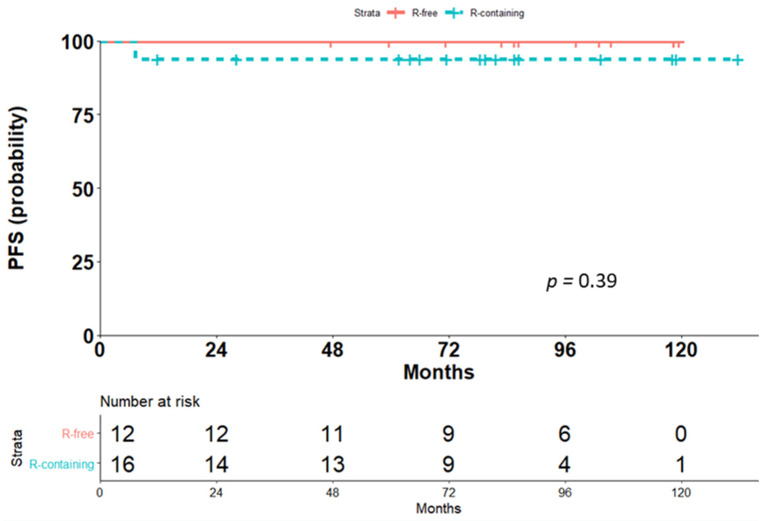
Comparison of overall survival (OS) for the two treatment groups, Rituximab-containing (R-containing) and Rituximab-free (R-free).

**Table 1 cancers-13-01760-t001:** Clinical features of the two cohorts of patients: historical cohort versus Rituximab-containing cohort.

Clinical Features	Historical Cohort (*n* = 12)	Rituximab-Containing Cohort (*n* = 16)	*p* Value
Age (median; range) years	33 (19–56)	36 (21–69)	0.30
Sex			
Males	11 (91.7)	16 (100)	0.88
Stage			
I	6 (50)	4 (25)	0.17
II	3 (25)	5 (31.25)	0.72
III	2 (16.7)	5 (31.25)	0.38
IV	1 (8.3)	2 (12.5)	0.72
B symptoms			
No	7 (70)	12 (75)	0.35
Yes	5 (30)	4 (25)	
Treatment groups			
Early favorable	4 (33.3)	6 (37.5)	0.82
Early unfavorable	5 (41.7)	3 (18.75)	0.18
Advanced-stage	3 (25)	7 (43.75)	0.30

Note: Unless otherwise indicated, data are number of patients, with percentage in parentheses.

**Table 2 cancers-13-01760-t002:** Risk-adapted strategies for historical cohort and Rituximab-containing cohort.

Treatment Groups	Historical Cohort	Rituximab-Containing Cohort
Early favorable (limited-stage)	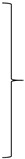	ABVD x 4 cycles + IF-RT (30 Gy)(limited-stage, *n* = 4;Intermediate-stage, *n* = 5)	Weekly induction Rituximab x 4 weeks + Rituximab maintenance (Rituximab once every 3 months for two years)(*n* = 6)
Early unfavorable (intermediate-stage)	ABVD x 4 cycles combined with monthly Rituximab x 4 months (on day 1 of each ABVD cycle)(*n* = 3)
Advanced-stage		ABVD x 6 cycles(*n* = 3)	ABVD x 6 cycles combined with Rituximab every two weeks for 6 months (on day 1 and 15 of each ABVD cycle) (*n* = 7)
ABVD [6]Doxorubicin: 25 mg/m^2^ IV days 1 and 15Bleomycin: 10 mg/m^2^ IV days 1 and 15Vinblastine: 6 mg/m^2^ IV days 1 and 15Dacarbazine: 375 mg/m^2^ IV days 1 and 15			
Rituximab [12]375 mg/m^2^ IV			
Risk factors according to the Germany Hodgkin Study Group:mediastinal mass larger than one-third of the maximum thoracic width;extra-nodal disease;elevated erythrocyte sedimentation rate (ESR): >50 mm/h without B symptoms or >30 mm/h with B symptoms (B symptoms: fever, night sweat, unexplained weight loss >10% over 6 months);Nodal areas: involvement of ≥3 nodal areas.Early favorable: clinical stage (CS) I-II without risk factors; early unfavorable: CS I-II with ≥1 risk factors; advanced-stage: CS stage IIB with large mediastinal mass and/or extranodal disease and CS III-IV.IF, involved-field; RT, radiotherapy; ABVD, Doxorubicin, Bleomycin, Vinblastine, Dacarbazine chemotherapy. IV, intravenous.

**Table 3 cancers-13-01760-t003:** Selected studies in early stage NLPHL, showing the relatively small number of patients treated with Rituximab-containing regimens.

Author, Year	Study Type	Reported Case	Stage	Treatment (Percentage of Patients)	PFS	OS	Median Follow-Up, Years	Adverse Events Grade ≥3 (Percentage of Patients)
Present Study, 2021	Retrospective	18	I–II	CMT (50)	89	100	7	Early + Late (55)
				RI + RM/R + CT (50)	100	100	7	Early (11)
Wilder, 2002 [16]	Retrospective	48	I–II	RT (100)	82%	83%	9	Late (7)
Feugier, 2004 [17]	Prospective	42	I–II	CMT (100)	80%	86%	15	nr
Nogovà, 2005 [18]	Retrospective	131	I	RT (69)	96%	100%	4	Early + late (7)
				CMT (31)	97%	100%		Early + late (49)
Wirth, 2005 [19]	Retrospective	202	I	RT (100)	82%	83%	15	Late (16)
Chen, 2010 [20]	Retrospective	113	I–II	CT (6)	14%	83%	11 years	nr
				RT (94)	77%	97%		
Savage, 2011 [21]	Retrospective	88	I–II	RT (36)	65%	84%	6	nr
				CMT (64)	91%	93%		
Advani, 2014 [12]	Prospective	14	I–II	RI (43)	42%	100%	9.5	nr
				RI + RM (57)	52%	100%	5	
Eichenauer, 2015 [22]	Retrospective	256	I	CMT (28)	88%	90%	8 years	Late (11)
				RT (61)	88%	98%		Late (6)
				RI (11)	81%	100%		Late (4)
Monteith, 2018 [23]	Prospective	29	I–II	CT	85%	95%	10	nr
				CMT	58%			
Alonso, 2018 [24]	Retrospective	1420	I–II	Observation (13)	nr	87%	4	nr
				RT (40)		93%		
				CT (22)		80%		
				CMT (25)		92%		
Pinnix, 2019 [25]	Retrospective	71	I–II	RT (51)	93%	100%	6	Late (8)
				CMT (41)	83%	88%		Late (17)
				CT (8)	67%	100%		nr
Borchmann, 2019 [26]	Retrospective	121	I–II	Observation (19)	65%	99%	6	nr
				RT (61)	94%			
				CMT (11)	81%			
				R+/-CMT+/-CT (7)	77%			
				CT (2)	nr			
Posthuma, 2019 [27]	Retrospective	441	I–II	CT (4)	nr	84%	8	nr
				CMT (11)		90%		
				RI (1)		100%		
				RT (57)		99%		
				Observation (27)		80%		
Eichenauer, 2020 [28]	Prospective	28	I	RI (100)	51%	91%	10	Late (11)
Binkley, 2020 [29]	Retrospective	559	I–II	RT (46)	91.1%	99.4%	5.5	Early * + late (5)
				CMT (33)	90.5%	99.4%		Early * + late (4)
				CT (8)	77.8%	97.9%		Early * + late (17)
				observation (7)	73.5%	89.8%		nr
				RI + RT (3)	80.8%	100%		nr
				RI (3)	38.5%	92.3%		nr

RT: radiotherapy; CMT: combined modality treatment; CT: chemotherapy; R: rituximab; I: induction; M: maintenance. * Early adverse events of grade 3 are reported together with grades 1 and 2; OS: overall survival.

**Table 4 cancers-13-01760-t004:** Selected studies in advanced-stage NLPHL, showing the relatively small number of patients treated with Rituximab-containing regimens.

Author, Year	Study Type	Reported Case	Ann-Arbor Stage	Treatment (Percentage of Patients)	Median Follow-Up, Years	PFS	OS	Adverse Events Grade ≥3 (Percentage of Patients)
Present Study, 2021	Retrospective	10	III–IV	CT (30)	2	33%	100%	Early (67)
				R + CT (70)	7	100%	85%	Early + late (43)
Xing, 2014 [30]	Retrospective	42	III–IV	CT (100)	11	82%	89%	Early * + late * (12.5)
Ames, 2015 [31]	Retrospective	8	III–IV	CT (100)	8	47%	nr	nr
Shankar, 2014 [32]	Retrospective	41	IIB–IV	R ±CT	7	78%	98%	nr
Fanale, 2017 [33]	Retrospective	22	III–IV	R + CT	7	86%	nr	nr
Prusila, 2018 [34]	Retrospective	9	III–IV	R + Bendamustine	3	100%	100%	nr
Borchmann, 2019 [26]	Retrospective	42	III–IV	Observation (33)	6	90%	100%	nr
				RT (2)		100%	100%	
				CT (33)		63%	97%	
				CT + R (32)		56%	100%	
Eichenauer, 2020 [35]	Retrospective	144	III–IV	CMT (100)	10	70%	87%	Early * + late (15.3)
Eichenauer, 2020 [36]	Prospective	84	III–IV	PET2 negative CT (63) PET2 positive CT (22.6)	5	90.2% 67.5%	nr	Late (12)
				PET2 positive R + CT (14.3)		73.3%		

RT: radiotherapy; CMT: combined modality treatment; CT: chemotherapy; R: rituximab. * Only adverse events leading to death are reported. PET2: positron emission tomography performed after two cycles of chemotherapy.

## Data Availability

The data presented in this study are available on request from the corresponding author. The data are not publicly available due to privacy restrictions.

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
