# Peer review of "Rituximab-Containing Risk-Adapted Treatment Strategy in Nodular Lymphocyte Predominant Hodgkin Lymphoma: 7-Years Follow-Up"

_cancers, 2021, doi:10.3390/cancers13081760_

Round 1
Reviewer 1 Report
The authors have substantially improved the manuscript. However, there are still some points that should be changed or revised:
- Rather use "NLPHL" instead of "NLP-HL"
- It is German Hodgkin Study Group (GHSG) not German Hodgkin Lymphoma Study Group. Please change.
- The term Rituximab-based is misleading since most patients had additional chemotherapy. Rituximab-containing would be more appropriate.
- Page 2 (line 67): Reference 2 and 7 do not fit here. Studies from the GHSG evaluating rituximab are: Eichenauer et al., Blood, 2011 (first-line treatment stage IA NLPHL), Eichenauer et al., Leukemia, 2020 (long-term follow-up of this study) and Schulz et al., Blood, 2008 (rituximab in relapsed NLPHL). Please revise.
- Page 4 (figure 1): Please delete the word noncontiguous. This is not the GHSG risk group allocation (advanced stages also include stage IIB with risk factors large mediastinal mass and/or extranodal disease)
- Table 3: References 9 and 23 report the same study. Maybe reference 9 can be deleted and only reference 23 is used (long-term follow-up of a prospective study). Reference 29 only reports patients with relapsed NLPHL retrospectively. Does this analysis fit here?
- Page 8 (line 197): Please also or solely discuss the follow-up analysis of the GHSG study evaluating rituximab in stage IA patients. At 10 years, 50% of patients had relapsed. This contradicts with the observation of the authors. On addition the 2 deaths observed in this study using rituximab alone were also due to second malignancies.
- Page 8 (line 217): This is very strong statement made on the basis of 6 patients treated at a single institution. In the reviewer´s opinion, ritximab can be discussed in early-stage NLPHL but there is certainly more evidence for the use of limited-field RT alone. Please attenuate the statement.
- Generally, it has to be pointed out even more than in the current version that only 16 patients treated with rituximab or rituximab plus chemotherapy were taken into account for the present analysis and that subgroups thus are very small.
Reviewer 2 Report
The authors compare "over all survival" and "progression-free survival" of Nodular lymphocyte-predominant Hodgkin lymphoma patients treated with Rituximab-based therapy or ABVD chemotherapy followed by radiotherapy. Table 3 and 4 summaries previous studies with and without Rituximab-based therapy. The presentation could be improved by adding a column providing the most relevant conclusion from each study. If there are few controversies then some general statements could be added in the main text. Currently, it is not transparent how many of the previous studies concur with the authors' conclusion. The authors' opinion on future perspectives could be improved in discussion.
Minor:
1. L. 58-63: very long sentence consider rephrase to improve readability. There are a few more sentences like this in the manuscript.
2.
L. 359 AU: "long rank test" suggestion "logrank test"
Reviewer 3 Report
thanks for considering my recommendations
Author Response
Thanks for you suggestion
Reviewer 4 Report
Even though the authors improved the paper with slight modification; the most important issues remain unchanged.
Author Response
Thank you for your comments
Round 2
Reviewer 1 Report
The authors worked on the issues raised by the reviewer sufficiently. There are only two points left that should be addressed:
1) The contents of reference 36 (abstract from EHA on PET-2-guided escalated BEACOPP in advanced NLPHL) has recently been published as full text (Eichenauer et al, Ann Oncol, 2021). Please add the full text publication as reference and also briefly discuss the results in connection with the use of rituximab (R-BEACOPP does not appear to result in better outcomes than standard escalated BEACOPP alone in PET-2-positive patients with advanced NLPHL).
2) Please check the reference style (for some references a DOI is indicated, for some not) and some references are not fully correct (e.g. reference 8 which has not been published in the Blood proceedings but in Blood)
Reviewer 4 Report
No further comment
Author Response
Thanks for your comments
This manuscript is a resubmission of an earlier submission. The following is a list of the peer review reports and author responses from that submission.
Round 1
Reviewer 1 Report
This era-to-era comparison of patients with newly diagnosed NLPHL either treated with rituximab alone or rituximab in combination with chemotherapy and/or RT or conventional treatment not comprising an anti-CD20 antibody generally addresses a topic of interest. However, there are some major points that should be addressed or revised:
- Did the median follow-up differ between both treatment groups? Please indicate the median follow-up times seperately for both groups. If the follow-up times differ between both groups, this should be discussed as a possible additional explanantion for the detected PFS differences.
- Did the patients´ biopsies undergo expert review? This information should be added.
- There might also have been some improvements in terms of diagnostics over time allowing a more accurate diagnosis of NLPHL in the group of patients receiving rituximab-containing treatment. Have there been cases with features of TCRBCL (elevated LDH, splenic involvement,...) especially in the group of patients not receiving rituximab-containing treatment?
- Are the 7-year PFS data reported? If yes, add the "7" in the text.
- In the reviewer´s opinion, rituximab maintenance does explain the 100% PFS. All other studies using rituximab alone did show much lower PFS rates (also those applying rituximab maintenance). How long was the follow-up among patients receiving rituximab alone?
- Do you think rituximab alone should also be applied in patients with stage IA NLPHL without risk factors? These patients do very well with limited-field RT alone. What about resection alone in case of a single affected lymph node and CR after lymph node resection (see Appel et al., J Clin Oncol)?
- Please explain why ABVD (plus rituximab) represents the standard protocol for the treatment of NLPHL in your hands? R-CHOP is possibly more appropriate in many cases (higher dose of alkylating agents which appear to be of major importance in NLPHL). What about the role of BR?
- If using rituxiamb in combination with chemotherapy, wouldn´t it be possible to omit bleomycin or reduce the dose of other agents included in this protocol? Please discuss the possibility to reduce the burden of conventional chemotherapy if rituximab is used as part of the treatment.
- The overall number of patients does not allow final conclusions. This should be mentioned more clearly. Maybe data from registries or pooled data from larger study groups or insitutions can shed more light on the role of rituximab in the first-line treatment of NLPHL.
- Language requires improvement.
Author Response
Reviewer 1
This era-to-era comparison of patients with newly diagnosed NLPHL either treated with rituximab alone or rituximab in combination with chemotherapy and/or RT or conventional treatment not comprising an anti-CD20 antibody generally addresses a topic of interest. However, there are some major points that should be addressed or revised:
- Did the median follow-up differ between both treatment groups? Please indicate the median follow-up times separately for both groups. If the follow-up times differ between both groups, this should be discussed as a possible additional explanation for the detected PFS differences.
We have now report in the Result section the median follow-up times separately for both groups, and the p-value for t-student tests showed no differences between treatment group follow-up mean times.
- Did the patients´ biopsies undergo expert review? This information should be added.
We have now specified in the Material and Methods section who were the expert hematopathologists reviewing the biopsies.
- There might also have been some improvements in terms of diagnostics over time allowing a more accurate diagnosis of NLPHL in the group of patients receiving rituximab-containing treatment. Have there been cases with features of TCRBCL (elevated LDH, splenic involvement,...) especially in the group of patients not receiving rituximab-containing treatment?
After reviewing process by the expert hematopathologists (as underlined in the Material and Methods section), no cases of TCRBCL were recorded.
- Are the 7-year PFS data reported? If yes, add the "7" in the text.
We have now specified in the Results section that the 7-year PFS was reported.
- In the reviewer´s opinion, rituximab maintenance does explain the 100% PFS. All other studies using rituximab alone did show much lower PFS rates (also those applying rituximab maintenance). How long was the follow-up among patients receiving rituximab alone?
We have now reported in the Discussion section the median follow-up with range in this subset of patients with early-stage disease without risk factors treated with rituximab induction+maintenance.
- Do you think rituximab alone should also be applied in patients with stage IA NLPHL without risk factors? These patients do very well with limited-field RT alone. What about resection alone in case of a single affected lymph node and CR after lymph node resection (see Appel et al., J Clin Oncol)?
Since treatment-related toxicities (including cytotoxic agents and/or radiotherapy) have been a major cause of mortality and morbidity in NLP-HL, in the Rituximab-era we preferred to use a less toxic treatment consisting of anti-CD20 antibodies alone in this setting of patients without risk factors. Lymph node resection alone wasn’t a treatment strategy in our Institution, because it was considered a therapeutic option in pediatric population.
- Please explain why ABVD (plus rituximab) represents the standard protocol for the treatment of NLPHL in your hands? R-CHOP is possibly more appropriate in many cases (higher dose of alkylating agents which appear to be of major importance in NLPHL). What about the role of BR?
We have briefly reported in the Discussion section the role of immunochemotherapy regimens used as alternative to ABVD.
- If using rituxiamb in combination with chemotherapy, wouldn´t it be possible to omit bleomycin or reduce the dose of other agents included in this protocol? Please discuss the possibility to reduce the burden of conventional chemotherapy if rituximab is used as part of the treatment.
We believe the Reviewer has identified an important point. We have now reported in the discussion section the possibility of omitting bleomycin when we added rituximab.
- The overall number of patients does not allow final conclusions. This should be mentioned more clearly. Maybe data from registries or pooled data from larger study groups or insitutions can shed more light on the role of rituximab in the first-line treatment of NLPHL.
Now, we argue this point in the discussion section.
- Language requires improvement.
Our manuscript was reviewed by a native English speaker.
Reviewer 2 Report
Review of "Rituximab-based risk-adapted treatment strategy in nodular lymphocyte-predominant Hodgkin lymphoma: 7-years follow-up"
The authors evaluate rituximab treatment of Nodular lymphocyte-predominant Hodgkin lymphoma. The find that rituximab treatment can limit the use of cytotoxic agent and/or radiotherapy related damage. Rituximab inclusion in advanced-stage therapeutic strategy appears to improve progression-free survival. The number of cases after grouping into cancer stages is the main limitation in the study. However, the authors discuss this limitation and defend it with the rarity of Nodular lymphocyte-predominant Hodgkin lymphoma. The study should have an interest for other clinicians as a study to follow up and reflect on.
Minor:
- Table 3 and 4 can be updated with the current study to facilitate comparison.
- Recent reviews on the subject have included additional studies on rituximab which were not mentioned by the authors e.g. British Journal of Haematology, 2019, 184, 17–29
- The abbreviation "R" for rituximab. Consider to just use rituximab. R is the standard abbreviation for many words.
- The reference to table 3 and 4 could also be introduced early in the discussion.
Author Response
Review 2
Comments and Suggestions for Authors
Review of "Rituximab-based risk-adapted treatment strategy in nodular lymphocyte-predominant Hodgkin lymphoma: 7-years follow-up"
The authors evaluate rituximab treatment of Nodular lymphocyte-predominant Hodgkin lymphoma. The find that rituximab treatment can limit the use of cytotoxic agent and/or radiotherapy related damage. Rituximab inclusion in advanced-stage therapeutic strategy appears to improve progression-free survival. The number of cases after grouping into cancer stages is the main limitation in the study. However, the authors discuss this limitation and defend it with the rarity of Nodular lymphocyte-predominant Hodgkin lymphoma. The study should have an interest for other clinicians as a study to follow up and reflect on.
Minor:
- Table 3 and 4 can be updated with the current study to facilitate comparison.
We have now reported in the first lines of both tables 3 and 4 data regarding the present study.
- Recent reviews on the subject have included additional studies on rituximab which were not mentioned by the authors e.g. British Journal of Haematology, 2019, 184, 17–29
We believe the Reviewer has identified an important point. However, we excluded the upper mentioned review because the authors included some studies reporting on pediatric patients; we decided to focalize our attention on adult population only.
- The abbreviation "R" for rituximab. Consider to just use rituximab. R is the standard abbreviation for many words.
Thank you for bringing this accidental error to our attention. We have now reported Rituximab instead of “R” in the main text, table and figure legends.
- The reference to table 3 and 4 could also be introduced early in the discussion.
We have now introduced in the Discussion section reference to tables 3 and 4, respectively at the head of the paragraph describing early-stage and advanced-stage treatment.
Reviewer 3 Report
The manuscript by Pugliese et al "Rituximab-based risk-adapted treatment strategy in nodular lymphocyte predominant Hodgkin lymphoma: 7-years follow-up" is a single institution study comparing 2 cohorts with nodular lymphocyte predominant Hodgkin lymphoma treated with rituximab in the latest era versus rituximab-free regimen in the historical era.
Line 92: Please separate the median follow up by eras.
Line 102: please describe the time and cause of death for the patient in the rituximab arm that appears to have died during therapy or shortly thereafter. Was it related to his rituximab therapy?
Table 2 and text: please explain if in the historical cohort all early stage patients were treated with 4 ABVD and RT. Also, please clarify if how many received RT, since in the table it says +RT, but in the methods part it says that they were treated with or without RT - what was the RT prescription based on?
Also in table 2 the risk classification is described, but this is obvious and appears to be buried. Please make it more clear that the risk classification was performed according to the GHSG risk factors and rather than give them letters, number them.
Methods: please describe the risk stratification and definitions in the text.
Line 242-243: did you get written informed consent for a retrospective study? how did you handle lost to follow up or the patient that died? or did the patients give informed consent for therapy, and this study was expect? It is unusual to get consent for a retrospective study.
Line 258-62. It appears that the risk stratification for the historical control was different if there was no early unfavorable, please clarify and if this is so, then I don't see a reason to have a figure comparing early unfavorable between both cohorts and given the small numbers it may make more sense to just have a figure for advanced stages and another one for early stages. Please clarify if all patients treated with only 4 cycles ABVD received IFRT or not.
Line 285-286: please clarify if long term effects are recorded prospectively or if they were gathered retrospectively. Normally CTCAE are used prospectively.
Given that it is not clear what the median follow up of the rituximab treated patients is and that many of them may still relapse, particularly if they are on maintenance rituximab, the conclusions of this study need to be relativized and soften.
I agree with the authors that their data contribute to the many other studies, most of which are retrospective in nature and adds to confirm that rtuximab likely is beneficial for patients with nLPHL, particularly those with advanced disease, but no absolute statements can be made with the data at hand.
Author Response
Review 3
Comments and Suggestions for Authors
The manuscript by Pugliese et al "Rituximab-based risk-adapted treatment strategy in nodular lymphocyte predominant Hodgkin lymphoma: 7-years follow-up" is a single institution study comparing 2 cohorts with nodular lymphocyte predominant Hodgkin lymphoma treated with rituximab in the latest era versus rituximab-free regimen in the historical era.
1) Line 92: Please separate the median follow up by eras.
In the Results section (sub-paragraph of “Outcome measures and survival”), we now specify the median follow-up for the entire study population and separately for the two treatment cohorts.
2) Line 102: please describe the time and cause of death for the patient in the rituximab arm that appears to have died during therapy or shortly thereafter. Was it related to his rituximab therapy?
For the patient died in the rituximab arm, we now describe the time and cause of death in the Results section and its relationship with bleomycin treatment in the Discussion section also.
3) Table 2 and text: please explain if in the historical cohort all early stage patients were treated with 4 ABVD and RT. Also, please clarify if how many received RT, since in the table it says +RT, but in the methods part it says that they were treated with or without RT - what was the RT prescription based on?
We now explain the RT employment in both the Table 2 and the text of the Material and Methods section (sub-paragraph of “Treatment”).
4) Also in table 2 the risk classification is described, but this is obvious and appears to be buried. Please make it more clear that the risk classification was performed according to the GHSG risk factors and rather than give them letters, number them.
In Table 2, we have now numbered the risk factors according to the GHLSG.
5) Methods: please describe the risk stratification and definitions in the text.
We now report in the Material and Methods section the risk stratification and its definition according to the GHSG.
6) Line 242-243: did you get written informed consent for a retrospective study? how did you handle lost to follow up or the patient that died? or did the patients give informed consent for therapy, and this study was expect? It is unusual to get consent for a retrospective study.
We believe the Reviewer has identified an important point. For all analyzed patients, we get informed consent at the time of NLP-HL diagnosis both in collecting clinical features in a data-base registry (experimental or scientific purposes) and for administering anti-lymphoma therapy.
7) Line 258-62. It appears that the risk stratification for the historical control was different if there was no early unfavorable, please clarify and if this is so, then I don't see a reason to have a figure comparing early unfavorable between both cohorts and given the small numbers it may make more sense to just have a figure for advanced stages and another one for early stages. Please clarify if all patients treated with only 4 cycles ABVD received IFRT or not.
We believe the Reviewer has identified an important point. Although the early favorable and unfavorable patients in the historical cohort received the same treatment, the patients in the rituximab cohort received different rituximab protocols (if they were early favorable or early unfavorable). Thus, we decided to differentiate the curves for these settings. Furthermore, we now explain the RT employment for early favorable and early unfavorable stages of the historical cohort in both the Table 2 and the Material and Methods section (sub-paragraph of “Treatment”).
8) Line 285-286: please clarify if long term effects are recorded prospectively or if they were gathered retrospectively. Normally CTCAE are used prospectively.
Thank you for bringing this to our attention. Although long-term side-effects were retrospectively recorded, we prefer to use CTCAE in the text to better classify their grade of severity.
9) Given that it is not clear what the median follow up of the rituximab treated patients is and that many of them may still relapse, particularly if they are on maintenance rituximab, the conclusions of this study need to be relativized and soften.
We believe the Reviewer has identified an important point. We now report in the Results section (sub-paragraph of “Outcome measures and survival”), the specific median follow-up of rituximab cohort and of historical cohort. Furthermore, we report in the Discussion section the median follow-up also of the early favorable patients treated with Rituximab induction for 4 weeks followed by Rituximab maintenance for two years. Finally, we softened our conclusions in agreement with the reported follow-up.
I agree with the authors that their data contribute to the many other studies, most of which are retrospective in nature and adds to confirm that rtuximab likely is beneficial for patients with nLPHL, particularly those with advanced disease, but no absolute statements can be made with the data at hand.
Reviewer 4 Report
Pugliese et al report here a series of 16 NLP-HL treated in the R era (2007-2018, treated with R single agent, or R-chimo) and compare their outcome to a control group of 12 cases treated w/o R (historical cohort, treated with ABVD +:- radiation).
Major comment:
- The study lack of novelty and does not add data to the existing literature.
- The size of the cohort, heterogeneity of the treatment / FU, and methodology issue (biases in patient selection) preclude from any interpretation.
- The precise aim or primary endpoint of the study is not clear to the reader: the author should focus on early stage or advanced NLP-HL and increase the N of cases.
- Any subgroup analysis do not seem to be reasonable given the low N of cases: ie comparison based on stage and GHLSG group classification include too few cases.
- Figure 1: survival curves and p value including 3 and 5 patients does not really make sens
- Treatment related bias in the selection of the historical and R cohorts: All the patients in the historical cohort were treated with chimo, so the higher incidence of cytopenia versus R alone is expected.
- None of them were observed, whereas this strategy would be the one to compare to R single agent;
- as well, none of the cases received radiation alone in any cohort whereas this strategy is the mainstay of treatment in many institutions.
- Recent guidelines do not recommend the use of R single agent for early favorable NLP-HL, due to inferior outcomes compare to RT or combined modalities (cf Spiner et al, BJH 2018). The authors should discuss this aspect. If only 3 cases in this series received R maintenance, this is certainly not enough to draw any conclusion.
- All in all, it seems that the treatment modalities are not fully representative of NLP-HL guidelines.
- Table 2 and text should include the N of cases for the treatment strategies : how many patients were treated with R alone ? CT ?...
Author Response
Review 4
Comments and Suggestions for Authors
Pugliese et al report here a series of 16 NLP-HL treated in the R era (2007-2018, treated with R single agent, or R-chimo) and compare their outcome to a control group of 12 cases treated w/o R (historical cohort, treated with ABVD +:- radiation).
- Major comment:
The study lack of novelty and does not add data to the existing literature.
The size of the cohort, heterogeneity of the treatment / FU, and methodology issue (biases in patient selection) preclude from any interpretation.
The precise aim or primary endpoint of the study is not clear to the reader: the author should focus on early stage or advanced NLP-HL and increase the N of cases.
Any subgroup analysis do not seem to be reasonable given the low N of cases: ie comparison based on stage and GHLSG group classification include too few cases.
Figure 1: survival curves and p value including 3 and 5 patients does not really make sens
Treatment related bias in the selection of the historical and R cohorts: All the patients in the historical cohort were treated with chimo, so the higher incidence of cytopenia versus R alone is expected.
None of them were observed, whereas this strategy would be the one to compare to R single agent;
as well, none of the cases received radiation alone in any cohort whereas this strategy is the mainstay of treatment in many institutions.
Recent guidelines do not recommend the use of R single agent for early favorable NLP-HL, due to inferior outcomes compare to RT or combined modalities (cf Spiner et al, BJH 2018). The authors should discuss this aspect. If only 3 cases in this series received R maintenance, this is certainly not enough to draw any conclusion.
All in all, it seems that the treatment modalities are not fully representative of NLP-HL guidelines.
Table 2 and text should include the N of cases for the treatment strategies: how many patients were treated with R alone ? CT ?...
We believe the Reviewer has identified important points. Now, we have improved the manuscript accordingly.
We are aware that absolute statements cannot be made due to small sample size and population heterogeneity, but we believe that our data can contribute to the current literature, most of which is retrospective and also based on small number of patients, and confirm that most likely Rituximab is beneficial for patients with NLP-HL, particularly those with advanced disease.
Due to the NLP-HL rarity, it is difficult to provide a larger sample size. We, however,
decided to focus on management and treatment results based on risk stratification to suggest a possible treatment that need to be necessarily confirmed in a larger population.
We treated patients in the historical cohort according to Italian guideline of that period (Brusamolino et al 2009), so RT alone wasn’t a treatment option in our Institution.
Recent guidelines do not recommend the use of Rituximab single agent for early favorable NLP-HL, but these data are based on a small numbers and also the role of Rituximab maintenance was not explored (Spinner et al 2018).
In Table 2, we report the number of cases for each treatment groups.